The classification of EEG-based winking signals: a transfer learning and random forest pipeline

Mahendra Kumar Jothi Letchumy 1
Rashid Mamunur 2
Muazu Musa Rabiu 3
Mohd Razman Mohd Azraai 1
http://orcid.org/0000-0002-0625-2327 Sulaiman Norizam 2
Jailani Rozita 4
http://orcid.org/0000-0002-3094-5596 P.P. Abdul Majeed Anwar 1 5 amajeed@ump.edu.my
1 Innovative Manufacturing, Mechatronics and Sports Laboratory, Faculty of Manufacturing and Mechatronics Engineering Technology, Universiti Malaysia Pahang , Pekan, Pahang Darul Makmur , Malaysia
2 Faculty of Electrical and Electronics Engineering Technology, Universiti Malaysia Pahang , Pekan, Pahang , Malaysia
3 Centre for Fundamental and Liberal Education, Universiti Malaysia Terengganu , Kuala Nerus, Terengganu , Malaysia
4 Faculty of Electrical Engineering, Universiti Teknologi MARA , Shah Alam, Selangor , Malaysia
5 Centre for Software Development & Integrated Computing, Universiti Malaysia Pahang , Gambang , Malaysia
Maggi Laura
Electronic publication date: 2021 Mar 31
Publication date: 2021
Volume: 9
Electronic Location ID: e11182
Received 2020 Oct 13; Accepted 2021 Mar 8
Copyright: © 2021 Mahendra Kumar et al.
Copyright year: 2021
Copyright holder: Mahendra Kumar et al.
License: This is an open access article distributed under the terms of the Creative Commons Attribution License, which permits unrestricted use, distribution, reproduction and adaptation in any medium and for any purpose provided that it is properly attributed. For attribution, the original author(s), title, publication source (PeerJ) and either DOI or URL of the article must be cited.
License URL: https://creativecommons.org/licenses/by/4.0/

Keywords: Random forest, EEG, Winking, Continuous wavelet transform, Transfer learning

Funding: Universiti Malaysia Pahang RDU180321 This work was supported by Universiti Malaysia Pahang via grant number RDU180321. The funders had no role in study design, data collection and analysis, decision to publish, or preparation of the manuscript.

==============================
Brain Computer-Interface (BCI) technology plays a considerable role in the control of rehabilitation or peripheral devices for stroke patients. This is particularly due to their inability to control such devices from their inherent physical limitations after such an attack. More often than not, the control of such devices exploits electroencephalogram (EEG) signals. Nonetheless, it is worth noting that the extraction of the features and the classification of the signals is non-trivial for a successful BCI system. The use of Transfer Learning (TL) has been demonstrated to be a powerful tool in the extraction of essential features. However, the employment of such a method towards BCI applications, particularly in regard to EEG signals, are somewhat limited. The present study aims to evaluate the effectiveness of different TL models in extracting features for the classification of wink-based EEG signals. The extracted features are classified by means of fine-tuned Random Forest (RF) classifier. The raw EEG signals are transformed into a scalogram image via Continuous Wavelet Transform (CWT) before it was fed into the TL models, namely InceptionV3, Inception ResNetV2, Xception and MobileNet. The dataset was divided into training, validation, and test datasets, respectively, via a stratified ratio of 60:20:20. The hyperparameters of the RF models were optimised through the grid search approach, in which the five-fold cross-validation technique was adopted. The optimised RF classifier performance was compared with the conventional TL-based CNN classifier performance. It was demonstrated from the study that the best TL model identified is the Inception ResNetV2 along with an optimised RF pipeline, as it was able to yield a classification accuracy of 100% on both the training and validation dataset. Therefore, it could be established from the study that a comparable classification efficacy is attainable via the Inception ResNetV2 with an optimised RF pipeline. It is envisaged that the implementation of the proposed architecture to a BCI system would potentially facilitate post-stroke patients to lead a better life quality.

Introduction

Stroke is a type of neurological disease that is the third leading cause of death and one of the top ten causes of mortality in Malaysia. The Global Burden of Disease estimated that stroke could be the second leading cause of mortality in 2040 (Ganasegeran et al., 2019). Patients suffering from stroke are often left with long term impairments (Murray & Harrison, 2004). Almost all of the patients are the affected with various degree of neurological disorder, that is not limited to the weakening of limbs or speech impairments (Lawrence et al., 2001; Schweizer & MacDonald, 2014).

The consequences of the impairments of the limbs are the restriction of the ability to perform rudimentary activities of daily living (ADL) (Norris, Allotey & Barrett, 2012). However, rehabilitation plays a vital role in the recovery process, facilitating patients to regain their ability to be independent. Hitherto, Brain-Computer Interface (BCI) has paved its way as one of the leading technologies for rehabilitation. A BCI system essentially provides communication between the human brain signal and external devices (Vaughan, 2003; Shih, Krusienski & Wolpaw, 2012; Lin & Hsieh, 2016). It is important to note that a successful BCI primarily has two main requirements, viz. a set of suitable electroencephalogram (EEG) features and an efficient machine-learning algorithm to classify the extracted features.

Related works

Over the last decade, active research has been carried out on the various feature extraction and classification techniques for EEG signals (Wang et al., 2015; Salgado Patrón & Barrera, 2016; Schwarz et al., 2018; Chronopoulou, Baziotis & Potamianos, 2019; Rodrigues, Jutten & Congedo, 2019). A pre-trained convolution neural networks (CNN) (a variation of the Transfer Learning model) was investigated to improve the BCI-system usability of a driving system that utilises EEG signals (Shalash, 2019). Online datasets were used in the research which was collected in a controlled lab environment through Neuro-scan data acquisition equipment with 30 effective channels and two reference electrodes. The collected EEG signals were converted into spectrogram images through the Short-Time Fourier Transform (STFT) algorithm. The converted images were implemented into the Alexnet TL model, which was trained via Adam optimiser with an initial rate of 0.0001. The datasets were divided into two separate datasets, i.e., training and testing with a ratio of 70:30. The results obtained showed that T3 and FP1 channels could yield reasonably high classification accuracy (CA) of 91% and 90%, respectively. It is evident from the study that TL facilitates the feature extraction process.

The detection of eye blinking from EEG signals was investigated by Domrös et al. (2013). The intentional eye-blink EEG signals were collected through a bio-radio device in the Biomedical Department Laboratory at Holy Spirit University. In the research, time-domain features, i.e., maximum amplitude, minimum amplitude and kurtosis, were extracted. The extracted features were then fed into the Gaussian Radial Basis Function (GRBF) model to classify the eye blink-EEG based signals accordingly. This pipeline was compared with other models, namely, the Multilayer Perceptron (MLP), Feed Forward Back Propagation, MLP-Cascade Forward Back Propagation (CFBP) and RBF Binary Classifier. The result showed that the GRBF classifier performed well based on the extracted time-domain features.

Rashid et al. (2020) investigated the classification of wink-based EEG signals. The features of the EEG signals were extracted through the Fast Fourier Transform (FFT) and sample range methods (Rashid et al., 2020). The FFT algorithm was utilised to transform the EEG signals into frequency domain features. The extracted features were implemented into several different classical machine learning classifiers, namely Linear Discriminant Analysis (LDA), Support Vector Machine (SVM), and k-Nearest Neighbor (kNN). The results showed that LDA performed better than the other two classifiers with a CA of 83.3% and 80% for the train and test dataset, respectively, through the FFT features. Conversely, based on the sample range features, identical CA was obtained through both SVM and kNN. models, i.e., 98.9% and 96.7% for the test and train dataset, accordingly. The LDA recorded a lower CA than the aforesaid classifiers based on the sample range features; nonetheless, the CA was significantly higher than the FFT feature extraction technique.

A driver fatigue classification system through the use of TL models and single-channel EEG signals was investigated by Shalash (2019). The proposed pipeline was evaluated on the online dataset obtained from Min, Wang & Hu (2017) that was downsampled from 1,000 to 500 samples. The downsampled signals were converted into spectrogram images through Short Time Fourier Transform (STFT). A total number of 3,440 spectrogram images were generated from each channel. The features of the images were extracted via the AlexNet Transfer Learning (TL) model. The AlexNet model was set to an initial rate value of 0.0001 with a batch size of 32 and the decaying gradient factor of 0.7. The highest classification was obtained through signals obtained through channels T3 and FP1, with a CA of 91% and 90%, respectively, suggesting the efficacy of spectrogram transformed signals and the TL pipeline towards driver fatigue classification.

Kant et al. (2020) utilised a Continuous Wavelet Transform (CWT) algorithm for the classification process of Motor Imagery (MI) EEG signals. Different TL models with tuned fully connected layers were evaluated in classifying the EEG signals. The dataset utilised was the Dataset III of BCI competition 2003, which consists of MI signal of left-hand and right-hand movements. The signals were pre-processed via a bandpass filter between the frequency range of 8 Hz and 30 Hz. The filtered signals were converted into CWT scalogram images. The scalogram images then were fed into TL models such as VGG19, AlexNet, VGG16, SqueezeNet, ResNet50, GoogleNet, DenseNet201, ResNet18, and ResNet101. It was shown from the study that the proposed pipeline obtained a CA of 95.71% and was demonstrated to be the highest CA achieved compared to other reported studies that utilise the same dataset with different classification approaches.

Wang et al. (2020) investigated emotion recognition through the use of Electrode Frequency Distribution Maps (EFDM) via Short-Time Fourier Transform (STFT). The authors utilised SJTU Emotion EEG Dataset (SEED) to study the proposed pipeline. The SEED consists of emotion actions which are positive, negative and neutral feelings. Whereas, Dataset for Emotion Analysis through Physiological (DEAP) was used to carry out cross dataset classification. The digital signals from both the dataset were converted into spectrogram images, and the classification was carried out through a pre-trained CNN classifier. The Principle Component Analysis was executed to reduce the dimension of the features vectors that were generated through the STFT algorithm. The cross emotional dataset classification obtained CA less than 40% without the implementation of the TL model. Conversely, the use of the TL model managed to obtain a CA of 96.60%, further suggesting the superiority of the TL-based model in classifying emotion-based EEG signals.

Therefore, the present study focuses on the implementation of a number of pre-trained CNN models (herein known as a variation of “Transfer Learning” models) to extract the features of the wink-based EEG signals. A conventional machine learning model, namely Random Forest, is implemented along with the Transfer Learning models to classify the extracted features. It is worth noting that such a pipeline has yet been investigated with regards to wink-based EEG signals. It is worth noting that this is the first study to investigate such a machine learning pipeline with regards to wink-based EEG signals. The performance of the different Transfer Learning models in feature extraction that will be classified through an optimised RF classifier as well as tuned fully connected layers (herein known as conventional CNN) shall be appraised. It is anticipated that the suggested pipeline could be implemented into a BCI assistive-technology and promote a better quality of life for post-stroke patients.

Methodology

The classification of the EEG signals consists of four main steps, viz. signal collection, pre-processing, feature extraction and classification, respectively. A five-channel Emotiv Insight EEG device was used to collect the wink-based EEG signals (Heunis, 2016). The position of the channels is placed according to the International 10–20 system, and the channels are placed at node AF3, AF4, T7, T8 and Pz.

The wink-based EEG signals were collected from five healthy subjects aged between 22 and 29 years old. The five subjects consist of three males and two females. The subjects that were chosen was ascertained not to have any medical problem and have normal vision. Moreover, it is worth noting that the subjects did not have any history of neurological diseases. A written informed consent form was received from the subjects who participated in the present study. The subjects were told to relax and sit on an ergonomic chair in a circumscribed room which is located at the Faculty of Electrical and Electronics Engineering Technology, University Malaysia Pahang. These steps were taken to avoid external signals to be recorded. The ethical approval for this study was obtained through an institutional research ethics committee provided by Universiti Kebangsaan Malaysia (FF-2013-327).

The subjects were instructed through a slide show displayed on LCD. The experiment paradigm shown in Fig. 1 were used to collect the required signals. The collection starts with the first five seconds of a resting-state, followed by winking action for the next five second. This step is continued to obtain six samples of winking signals. Left and Right winking action were run separately, and both of them were recorded for 60 s (1 min).

Figure 1 The experiment paradigm for EEG signal acquisition.

Continuous wavelet transform

Continuous Wavelet Transform (CWT) is the representation of the time-frequency domain of a set of signals collected. CWT is one of the most effective methods used in medical fields, which consists of non-stationary signals such as EEG, electromyography (EMG) or electrocardiogram (ECG), amongst others. The resolution represented through the CWT algorithm has been reported to advantageous due to the utilisation of the small scale of high frequencies and large scale of low frequencies (Türk & Özerdem, 2019). Moreover, it has also been reported to provide a better representation of the arrangement of the frequency domain features as compared to Fourier Transforms. The mother wavelet that was utilised in this research is the Morlet Wavelet. Morlet wavelet is the multiplication of the complex exponential and Gaussian window. It is worth noting that the Morlet wavelet method is widely used in the medical field, which consists of non-stationary signals (Qassim et al., 2012). The Morlet algorithm gives an intuitive association between frequency and time domain to distinguish the signals acquired via Fourier Transform.

Feature extraction: transfer learning

In the present investigation, a variation of Transfer Learning (TL) models that are governed by pre-trained Convolutional Neural Network (CNN) models is employed. Figure 2 depicts the architecture of the CNN pipeline and Pre-Trained CNN with Machine Learning (ML) pipeline ( also known as the TL- Classical Classifier pipeline). A Convolutional Neural Network (CNN) model is made up of four layers, which are convolutional layers, activation layer, pooling layers and fully connected layers. The input images will be fed into the convolutional layer, which consists of a filter that could slide over the input images and perform dot product operation, creating an activation map typically known as the feature map. The next layer will be the activation which consists of the ReLU function that performs a non-linear operation. Conversely, the pooling layer decreases the dimension of the activation map into a smaller dimension but preserves the significant features.

Figure 2 The architecture of CNN pipeline and TL-classical classifier pipeline.

It could be seen from Fig. 2 that the convolutional layer is frozen as it is solely used to extract the features from the transformed signal. Such TL models are widely used in computer vision amongst other fields, primarily owing to its ease in the CNN model development, especially omitting the notion of building the model from scratch (as pre-trained models are used) and hence reduces the model development time (Amanpour & Erfanian, 2013; Chronopoulou, Baziotis & Potamianos, 2019). This approach is also rather advantageous in bioinformatic related domains as data is often scarce, and it has been demonstrated in the literature such an approach is able to work with limited dataset. Table 1 illustrates the TL models and the parameters that were implemented in the present study. It is worth mentioning that the TL models used in the study are used only for feature extraction where only the convolutional layers are exploited. In contrast to a full pre-trained CNN model, the fully connected layers (dense layers) are replaced with a traditional machine learning classifier in the study, which in this case, the Random Forest classifier is employed.

Table 1 List of TL models and its respective parameters implemented in this research.

No.	Transfer learning models	Flatten size	Input image size	
1	Inception V3	8 × 8 × 2048	299 × 299	
2	Inception ResNetV2	8 × 8 × 1536	299 × 299	
3	Xception	10 × 10 × 2048	299 × 299	
4	MobileNet	7 × 7 × 1024	224 × 224	

Classifiers: random forest

Random Forest (RF), also known as Random Decision Forests, is a supervised machine learning algorithm that evolved through the ensemble of multiple Decision Tree classifiers. It is also known as one of the many bagging-type ensemble classifiers. The combination of a few decision trees to mitigates the notion between the variance and bias, which in turn reduces the possibility of overfitting. It is worth noting that the RF classifier has been widely used in many different medical oriented types of research (Cherrat, Alaoui & Bouzahir, 2020; Tabares-Soto et al., 2020). The RF hyperparameters evaluated in this study are the number of trees (n_estimators), depth of the trees (max_depth), and the measurement of the splitting quality (criterion). The hyperparameters of the RF models were tuned via the grid search algorithm through the five-fold cross-validation technique. Table 2 lists the hyperparameter values of RF classifiers appraised. A total of 98 RF models were investigated in this research for four different TL models (conclusively, a total of 392 TL pipelines were evaluated) towards its efficacy in classifying the wink-based EEG signals. Figure 3 depicts the complete pipeline developed in this study. The developed pipelines (different TL models with their associated optimised RF models) was analysed and evaluated using a Python IDE, specifically Spyder 3.7.

Figure 3 The complete TL pipeline.

Table 2 Hyperparameter of the RF models evaluated.

No.	Hyperparameters	Hyperparameter values	
1	n_estimator	10, 20, 30, 40, 50, 60, 70	
2	Max_depth	10, 20, 30, 40, 50, 60, 70	
3	Criterion	Gini and Entropy	

The optimised classical RF classifier with TL pipeline performance was compared against the TL with fully connected layers (herein referred to CNN for brevity) to provide a baseline comparison between the pipelines. The features were extracted as per the aforesaid TL models. The fully connected layers consist of two hidden layers. The first hidden layer consists of 50 hidden neurones with a ReLU activation function. Conversely, the final hidden layer consists of three neurones with a Softmax activation function which corresponds to the classes that we intend to investigate in the present study, i.e., left-wink, right-wink and no-wink. The Adam algorithm has been implemented as the optimiser to reduce the loss function. The proposed CNN architecture includes a dropout value of 0.5 with a batch normalisation size of 10, which plays a significant role in enhancing the classification accuracy. Dropout works as the “temporarily discarding” some neurone nodes with a certain probability during the training of a deep network. This process decreases the risk of overfitting and improves the generalisation ability of the model. The epoch value was set to 50 in this study.

Performance evaluation

The confusion matrix is one of the most straightforward and simplest measures used to determine model consistency and correctness (Sokolova & Lapalme, 2009; Flach, 2019). The classification models employed in this study are assessed by means of classification accuracy (CA), precision, recall, F1-score, specificity and Receiver Operating Characteristics (ROC) curve. The accuracy is simply the ratio between the number of accurately predicted observations and the total number of observations. The precision measures the percentage of correct positive forecasts over the cumulative number of positive forecasts. The recall (often known as sensitivity) is the number of true positive predictions divided by the sum of true positives as well as the false negatives (Vijay Anand & Shantha Selvakumari, 2019). The F1-score discloses the balance between the recall and the precision values. In contrast, specificity is the proportion of actual negative values, which is predicted as the true negative. The ROC curve is the measure of separability between the classes in a dataset.

Experimental results and discussion

The wink-based EEG signals were extracted through the single-channel Emotiv device at the sampling rate of 128Hz. The digital signals were then converted into scalogram via CWT. The images were divided into three groups of datasets: training, validation, and test datasets, through a stratified ratio of 60:20:20. The stratification ensures that the datasets are equally divided amongst the evaluated classes. The images were then fed into the TL models and classified through CNN and optimised RF models. Figs. 4 and 5 depicts the raw and scalogram transformed images of the wink classes.

Figure 4 Plot of raw EEG signal (A) left-wink (B) right-wink and (C) no-wink.

Figure 5 Scalogram of (A) left wink (B) right wink (C) no wink.

The performance of the pipelines was evaluated through stratified divided datasets, which are training, validation and test datasets, respectively. Figure 6 depicts the CA obtained through the training and test datasets of the TL-CNN pipeline. Through the bar chart, it could be observed that a train CA of 100% was obtained via Inception ResNetV2 and MobileNet pipelines. Nonetheless, as for the test CA, the Inception ResNetV2 and MobileNet obtained 94% and 78%, respectively. Therefore, it is apparent that the Inception ResNetV2 TL works well with the tuned fully connected layer of the CNN model. The performance measures obtained through the test dataset for all the four pipelines evaluated are listed in Table 3.

Figure 6 Classification accuracy of TL-CNN pipelines.

Table 3 Performance measures of test datasets obtained via TL-CNN pipelines.

	Class	Precision	Recall	F1-score	Specificity	CA	
Performance measures obtained through inceptionV3 pipeline	
Left winking	0	1.00	1.00	1.00	1.00	1.00	
Right winking	1	1.00	1.00	1.00	1.00	
No winking	2	1.00	1.00	1.00	1.00	
Performance measures obtained through inception ResNetV2 pipeline	
Left winking	0	0.86	1.00	0.92	0.14	0.94	
Right winking	1	1.00	1.00	1.00	0.00	
No winking	2	1.00	0.83	0.91	0.17	
Performance measures obtained through Xception pipeline	
Left winking	0	0.50	1.00	0.67	0.50	0.67	
Right winking	1	1.00	1.00	1.00	0.00	
No winking	2	0.00	0.00	0.00	1.00	
Performance measures obtained through MobileNet pipeline	
Left winking	0	1.00	0.33	0.50	0.77	0.78	
Right winking	1	1.00	1.00	1.00	0.00	
No winking	2	0.60	1.00	0.75	0.40	

Figure 7 illustrates the CA obtained from training and test datasets by means of the TL- optimised Random Forest (RF) classifier. It is evident from the figure that all the evaluated TL-optimised RF classifier pipeline could attain a CA of 100% on the training dataset. Nevertheless, upon further evaluation of the test dataset, it is obvious that the Inception ResNetV2 and Xception pipelines performed well in the classification of the investigated wink-based EEG signals.

Figure 7 Classification accuracy of optimised TL-RF pipeline.

Both pipelines, i.e., TL-CNN and TL-optimised RF, were evaluated on validation dataset as a form of robustness evaluation. The CA obtained through the validation dataset by both the models are illustrated in Fig. 8. It could be seen that the pipeline, which is made up of CWT-Inception ResNetV2-Optimised RF, performed exceptionally well in the classification of wink-based EEG signals as no misclassification transpired. The optimised RF model’s hyperparameters that yielded such a result are ten (10) trees, twenty (20) tree depth and the Gini criterion.

Figure 8 The classification accuracy obtained through validation dataset of the pipelines used in this research.

Table 4 tabulates the performance measure of the test dataset based on the Inception ResNetV2-RF pipeline. Figure 9 represents the ROC curve obtained for the validation dataset through the Inception ResNetV2 pipeline. Figure 10 illustrates the confusion matrix of the validation dataset in which 0, 1 and 2 represent the left, right and no wink classes. Through the confusion matrix, it can observed that there is no misclassification that transpired in the classification of the validation dataset.

Figure 9 ROC curve of validation dataset through inception ResNetV2.

Figure 10 Confusion matrix of validation dataset via inception ResNetV2.

Table 4 Performance measures obtained through test dataset via TL-RF pipelines.

	Class	Precision	Recall	F1-score	Specificity	CA	
Performance measures obtained through inception V3 pipeline	
Left winking	0	0.86	1.00	0.92	0.14	0.94	
Right winking	1	1.00	1.00	1.00	0.00	
No winking	2	1.00	0.83	0.91	0.17	
Performance measures obtained through inception ResNetV2 pipeline	
Left winking	0	1.00	1.00	1.00	1.00	1.00	
Right winking	1	1.00	1.00	1.00	1.00	
No winking	2	1.00	1.00	1.00	1.00	
Performance measures obtained through Xception pipeline	
Left winking	0	1.00	1.00	1.00	1.00	1.00	
Right winking	1	1.00	1.00	1.00	1.00	
No winking	2	1.00	1.00	1.00	1.00	
Performance measures obtained through MobileNet pipeline	
Left winking	0	0.86	1.00	0.92	0.14	0.94	
Right winking	1	1.00	1.00	1.00	0.00	
No winking	2	1.00	0.83	0.91	0.17	

The efficacy of pre-trained CNN models have been demonstrated in the literature; for instance, Kant et al. (2020) implemented a CWT algorithm to classify motor imagery signals by means of transfer learning models. The digital EEG signals were converted into two-dimensional scalogram images that were fed into different pre-trained CNN models such as AlexNet, VGG16 and VGG19 to recognise the motor imagery signals of the Left- and Right-hand movements. It was shown from the study that the employment of such a technique could achieve a CA of 97.06%. In a different study, CWT transformation has been utilised along with a pre-trained CNN model, SqueezeNet, to classify sleep stage based on EEG signals (Jadhav et al., 2020). It was demonstrated that the pipeline could yield exceptional CA. It is apparent that the conversion of signals via CWT could provide meaningful features to be extracted through the transfer learning approach, which was also demonstrated through the present study. It is also worth noting that with regards to the classification of wink-based EEG signals, the present study has shown that exceptional classification was achieved via the proposed approach and was shown to be better than that of results reported by Rashid et al. (2020).

Conclusion

It could be shown from the present investigation that the employment of Transfer Learning is a rather promising approach in improving the performance of EEG classification for BCI applications. Different stages of winking were converted into a spectrogram image through CWT. It has been demonstrated through the study that the Inception ResNetV2-Optimised RF could provide a reasonable classification of the wink-based EEG signals as compared to the other TL models evaluated. In addition, it is also worth noting that the role of hyperparameter tuning could not be simply overlooked as it could further improve the performance of the evaluated classifier, herein, the RF for the present investigation. Future works shall evaluate the performance of other forms of classical classifiers, for instance, Support Vector Machine and k-Nearest Neighbours, amongst others and its combination with the evaluated TL models on such classification.

Supplemental Information

Supplemental Information 1 No wink EEG signals for subject 1.

Click here for additional data file.

Supplemental Information 2 Right wink EEG signals for subject 1.

Click here for additional data file.

Supplemental Information 3 Left wink EEG signals for subject 1.

Click here for additional data file.

Supplemental Information 4 No wink EEG signals for subject 2.

Click here for additional data file.

Supplemental Information 5 Right wink EEG signals for subject 2.

Click here for additional data file.

Supplemental Information 6 Left wink EEG signals for subject 2.

Click here for additional data file.

Supplemental Information 7 Right wink EEG signals for subject 3.

Click here for additional data file.

Supplemental Information 8 No wink EEG signals for subject 3.

Click here for additional data file.

Supplemental Information 9 Left wink EEG signals for subject 3.

Click here for additional data file.

Supplemental Information 10 No wink EEG signals for subject 4.

Click here for additional data file.

Supplemental Information 11 Right wink EEG signals for subject 4.

Click here for additional data file.

Supplemental Information 12 Left wink EEG signals for subject 4.

Click here for additional data file.

Supplemental Information 13 No wink EEG signals for subject 5.

Click here for additional data file.

Supplemental Information 14 Right wink EEG signals for subject 5.

Click here for additional data file.

Supplemental Information 15 Left wink EEG signals for subject 5.

Click here for additional data file.

Supplemental Information 16 Code: TL + FCL.

Click here for additional data file.

Supplemental Information 17 Code: TL hybrid with RF.

Click here for additional data file.

Supplemental Information 18 Code to convert signals to Scalogram.

Click here for additional data file.

Additional Information and Declarations

Competing Interests

Author Contributions

Human Ethics

Data Availability

The authors declare that they have no competing interests.

Jothi Letchumy Mahendra Kumar performed the experiments, authored or reviewed drafts of the paper, and approved the final draft.

Mamunur Rashid analysed the data, prepared figures and/or tables, and approved the final draft.

Rabiu Muazu Musa performed the experiments, analysed the data, authored or reviewed drafts of the paper, and approved the final draft.

Mohd Azraai Mohd Razman analysed the data, prepared figures and/or tables, and approved the final draft.

Norizam Sulaiman analysed the data, prepared figures and/or tables, and approved the final draft.

Rozita Jailani conceived and designed the experiments, authored or reviewed drafts of the paper, and approved the final draft.

Anwar P.P. Abdul Majeed conceived and designed the experiments, authored or reviewed drafts of the paper, and approved the final draft.

The following information was supplied relating to ethical approvals (i.e., approving body and any reference numbers):

Universiti Kebangsaan Malaysia granted Ethical approval to carry out the study within its and associated facilities (Ethical Approval Reference: FF-2013-327).

The following information was supplied regarding data availability:

Data and script are available in the Supplemental Files.

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
