# Peer review of "The classification of EEG-based winking signals: a transfer learning and random forest pipeline"

_PeerJ, doi:10.7717/peerj.11182_

## Round 0.1 · original submission · Major Revisions

The reviewers raised important criticisms which they described in detail. The manuscript requires extensive revision in order to be re-evaluated. It is advisable to reply to all the points raised by the 2 reviewers exhaustively.

Reviewer 1 ·

Basic reporting

The article is largely unambiguous and with sufficient background. The hypothesis and results are convincing to a large extent. Nevertheless, the author might consider reviewing the following.

Consider changing the sentence in lines 68-70 to "Particularly, a successful BCI primarily has two main requirements including a set of suitable electroencephalogram (EEG) features and an efficient machine-learning algorithm to classify the extracted features."

Also note that Transfer Learning does not apply only to CNN models or computer vision as implied in lines 76 and 164. It can be applied to other deep learning architectures.

Change "myriad" in line 111 to "several" or "a number of".

Please modify the statement, "It is worth noting that such a pipeline has yet been investigated with regards to wink-based EEG signals" in lines 114-115 to something like this: "It is worth noting that this is the first study to investigate such a machine learning pipeline with regards to wink-based EEG signals".

Change "till" in line 131 to "and".

Please delete "to transpire" in line 184.

Correct the spelling for "Inception" spelled as "Ineption" in Figure 5.

Experimental design

The technical and ethical standard of the work is acceptable. However, the methods need to clarified.

i) For instance, the method of transfer learning used was not fully described.
Were the retained pretrained layers' weights frozen or not during training of the classifiers? What is the justification for the approach adopted? E.g., performance reasons?

Further, there other clarifications that need to be provided to facilitate replication of results.

ii) What is the justification for the addition of RF classifier in the classification pipeline?
Was the performance of the TL models investigated in exclusion of the RF classifier?
What is the performance of the different TL models in that scenario?

iii) More so, you might want to check the performance of the RF classifier alone (i.e., in exclusion of TL).

The performance values realized from (ii) and (iii) could serve as a reference to demonstrate the benefit of the methods (combining TL and RF) applied in this study.

iv) Please include specificity and ROC AUC metrics to the performance evaluation metrics. In particular, It would be nice to consider specificity alongside the sensitivity values except the classes are balanced in the datasets.

Validity of the findings

The conclusion is in line with the aim of the study. However, clarification of the issues raised in the experimental design can improve the significance of findings.

Reviewer 2 ·

Basic reporting

A critical concern in this manuscript is that the authors used a very poor English to present. There are a lot of grammatical errors and typos. It is hard to follow and capture the ideas with poor English. Therefore, the authors should have an extensive revision on the English. It is important if it can be checked by a native speaker or English editing service.
Literature review are weak. The authors should show more latest works on the different feature extraction and classification techniques for EEG signals system.
Quality of figures should be improved. Now resolution is low.

Experimental design

The authors should release source codes for replicate the results.
How many experts were used for labeling? How to decide the experts?
Random algorithm has been used in previous works for classification of EEG. Thus the authors should refer more works to attract broader readership.

Validity of the findings

The authors have not compared their performance results with the previous works on the same problem. Therefore, it is not enough to convince the impact of the study.
The authors should have some independent tests on the model.
More discussions need to be added.

Additional comments

No comment.

---

## Round 0.2 · Minor Revisions

Before acceptance, authors should have a careful revision of the English. The quality of figures should be also improved, as suggested by the first reviewer.

Reviewer 1 ·

Basic reporting

Thanks to the author.
The concerns raised in my previous review have been largely attended to.

Experimental design

No additional comments.

Validity of the findings

No additional comments.

Additional comments

No additional comments.

---

## Round 0.3 · accepted · Accept

The concerns raised by reviewers have been attended and the paper is suitable for publication.